# Micro- and Nano-Based Transdermal Delivery Systems of Photosensitizing Drugs for the Treatment of Cutaneous Malignancies

**DOI:** 10.3390/ph14080772

**Published:** 2021-08-06

**Authors:** Isabella Portugal, Sona Jain, Patrícia Severino, Ronny Priefer

**Affiliations:** 1Programa de Pós-Graduação em Biotecnologia Industrial, Universidade Tiradentes, Aracaju 49032-490, Brazil; isabella.martins@souunit.com.br (I.P.); sona.arun@souunit.com.br (S.J.); patricia.severino@souunit.com.br (P.S.); 2Massachusetts College of Pharmacy and Health Sciences, University, Boston, MA 02115, USA

**Keywords:** photodynamic therapy, drug delivery, transdermal, cutaneous, cancer

## Abstract

Photodynamic therapy is one of the more unique cancer treatment options available in today’s arsenal against this devastating disease. It has historically been explored in cutaneous lesions due to the possibility of focal/specific effects and minimization of adverse events. Advances in drug delivery have mostly been based on biomaterials, such as liposomal and hybrid lipoidal vesicles, nanoemulsions, microneedling, and laser-assisted photosensitizer delivery systems. This review summarizes the most promising approaches to enhancing the photosensitizers’ transdermal delivery efficacy for the photodynamic treatment for cutaneous pre-cancerous lesions and skin cancers. Additionally, discussions on strategies and advantages in these approaches, as well as summarized challenges, perspectives, and translational potential for future applications, will be discussed.

## 1. Introduction

In past decades, clinical demands for the utilization of photosensitizers (PSs) have increased with the advent of photodynamic therapy (PDT). PDT is an alternative therapy whereby an initial activation of a PS at a specific wavelength (visible light or near-infrared (NIR) leads to the formation of reactive oxygen species (ROS), which in turn induces cell death. Notably, PDT has been primarily applied in cancer therapy. The advantages of low light irradiation therapies have made it possible to widen the range of target diseases [1]. For example, the antimicrobial properties of PDT (aPDT) have been applied for the treatment of bacterial, fungal, parasitic, and viral infections [2]. PDT has also been explored for wound healing [3,4], immune-mediated cutaneous diseases [5], anesthetic purposes [6], and aesthetic applications [7].

Historically, phototherapy can trace its roots to Ancient Egyptian, Chinese, and Indian civilizations. Light combined with photosensitizing natural formulations were used to treat illnesses such as vitiligo, psoriasis, and skin cancer. In 1900, Raab and von Tappeiner described the phenomena of cell death induced by a combination of chemicals (acridine dye) and light on the protozoa *paramecia* (Figure 1) [8]. Further investigations gave birth to the revolutionary *Photodynamic Action*. The first biomedical use of this innovation was reported by Friedrich Meyer-Betz, in 1912, who self-injected hematoporphyrin, resulting in pain and swelling on light-exposed areas [9]. Subsequently, in 1961, Lipson et al. demonstrated that hematoporphyrin derivatives (HpD) accumulated in tumors and emitted fluorescence could be applied as a diagnostic tool [10].

Further, Dougherty et al. introduced PDT in the 1970s by observing complete mammary tumor remission in vivo using HpD combined with the red light [11]. A subsequent clinical study using 25 patients showed complete response in 98 out of 113 skin tumors, partial response in 13, and only two resistant tumors [12]. These findings led to the first approval of the PDT drug, Photofrin^®^, to treat bladder cancer in Canada in 1993 [13]. In 2001, Foscan^®^ became the first second-generation PS agent commercially available for PDT, especially in head-and-neck squamous cell carcinoma (HNSCC) treatment. Ultimately, in 2012, silicon phthalocyanine (Pc) 4 entered a phase I clinical trial as the first topical PDT agent **(**Figure 1**)** [14]. Currently, PDT is employed to treat a wide range of diseases, such as leishmaniasis, psoriasis, neovascular macular degeneration, cardiology, urology, immunology, ophthalmology, dentistry, and dermatology [15].

The distinctiveness of PDT relies on its highly selective mechanism of action. Utilizing two independently nontoxic components (i.e., a PS and light) to produce cytotoxicity within a tumor. There is a high level of tumor selectivity to PS due to increased tumor vasculature surface area, higher membrane permeability of cancer cells, and decreased lymphatic drainage [16]. An ideal PS remains inert until a light source is focused onto the intended area to “activate” the PS, boosting its selectivity over surrounding healthy tissues. Ultimately, a third intrinsic component for PDT, molecular oxygen, present in tissue’s extracellular and intracellular spaces, serves as the substrate for ROS formation. The generation of singlet oxygen and superoxide anions results in tumor cytotoxicity as both can directly react with and damage biomolecules such as lipids, proteins, and nucleic acids [17].

Several studies have demonstrated PDT as a viable treatment option against early-stage esophageal dysplasia, lung, HNSC, anal, bladder, peritoneal ovarian, and non-melanoma skin cancers (NMSC) [18]. Despite the encouraging clinical results of PDT, some PSs themselves have been reported to have prolonged skin phototoxicity, low lesion selectivity, hydrophobic nature, aggregation proneness, poor bioavailability, high-dose requirements, adverse side effects, off-targeting, and development of drug resistance [19,20,21]. The use of drug delivery systems (DDS) to overcome these shortcomings has been examined. In this context, PSs can be ideally delivered to therapeutic action sites while reducing adverse side effects [21]. Among DDS, the transdermal route stands out for dermatological applications. However, the inherent protective epidermic layer, the high molecular weight of some PSs (>500 Daltons), and extremes of polarity remain a challenge for crossing the skin barrier [22].

Several innovative transdermal delivery systems (tDDS) have recently become available to study photobiology, improve drug penetration, enhance site-specific delivery, and increase therapeutic efficiency. This review aims to summarize current approaches focused on boosting the transdermal delivery efficacy of PSs for treating cutaneous conditions.

## 2. Transdermal Drug Delivery of PDT Agents

The skin barrier is a lipid layer along which individual molecules can migrate through diffusion. Although skin has a large surface area, transdermal drug delivery is challenging since it acts as a formidable barrier. Molecules can enter through keratinocytes (the transcellular pathway), the lipid matrix occupying the intercellular spaces (the intercellular pathway), hair follicles, sebaceous glands, or sweat glands (the transappendageal pathway) [23]. The process is controlled mainly by the molecule’s permeant hydrophilicity, size, and hydrogen-bonding ability [24]. Consequently, substances with molecular weight lower than 600 Da, water/octanol partition coefficient (logP values) in the range of 1–3, and those with a low melting point can be more easily absorbed into the skin. In contrast, larger molecules and highly polar compounds cannot pass the cutaneous barrier [25].

From the penetrant perspective, the skin behaves like a mechanical nanoporous barrier perforated by many gap-like or quasi-semicircular pathways. Previous studies estimate an average passage-width range of 0.4 to 36 nm. Arguably, pore-size distribution is non-symmetrical, with a peak around 20 nm and a steeper decline on the transbarrier pathways’ low-sized side. Moreover, existing data demonstrate that the critical factor for hydrophilic entity motion through the skin is primarily molecular size and not molecular weight [26]. Thus, several parallel physiological paths between two cells increase the skin porosity, which is crucial for topical drug delivery.

One extensively investigated strategy to enhance drug delivery through the skin is by adding chemical enhancers such as fatty acids, surfactants, esters, alcohols, pyrrolidones, amines, amides, sulphoxides, terpenes, alkanes, and phospholipids. Conversely, mechanical, physical, and active transport techniques, such as microneedles (MNs), jet injectors, iontophoresis, ultrasound, electroporation, photomechanical waves, magnetophoresis, laser radiation, and skin abrasions, are also available to enhance skin penetration [25]. The use of chemical enhancers is restricted due to the limited permeability of macromolecules, while physical approaches are invasive and can damage the skin barrier properties (Figure 2). To overcome some limitations of chemical and physical enhancers, nanoparticles are being developed to improve the absorption and transdermal drug release in a controlled manner for a prolonged period.

The application of nanotechnology for conventional PSs has helped to overcome many shortcomings, particularly related to the hydrophobic nature of these compounds, which limits the penetration through the skin and cell membranes [20,27]. While hydrophobic PSs, such as phthalocyanines (Pcs) and chlorins, form aggregates in physiological solutions [28], hydrophilic PSs raise interaction problems with biological tissues. In this context, liposomes have been reported as suitable delivery systems for carrying both hydrophilic and hydrophobic PSs and leading to improved clinical applications [29]. For instance, the extensively studied PS, 5-aminolevulinic acid (ALA), is not internalized well by cells and has no specificity for diseased tissues [27]. Therefore, PSs associated with nanoparticles have evolved to avoid self-aggregation and improve delivery to the target site.

## 3. Chemical Approaches for Transdermal Delivery

### 3.1. Nanoemulsion (NE)

The term emulsion denotes a suspension of globules of a liquid in another liquid, stabilized by small molecules with polar and nonpolar ends that can intercalate into bilayers nominated amphipaths. On opened skin, emulsions typically break into separate oil and water phases, containing amphipaths at their respective saturating concentrations. This principle is useful for in situ generation of a saturated drug solution in oil or water under the constraint of the accompanying skin occlusion by the resulting fatty layer at the skin surface [26].

The addition of a unipolar fluid component (an oil or oleaginous ingredient) to a water–amphipath mixture can generate a NE. Nanosized oil droplets emerge from emulsions after a mechanical disruption or, in rare cases, thermal agitation of the system. The average droplet size of NEs has been broadly defined from 50 to 500 nm, but is typically 200 to 300 nm [30]. Fluctuations in all described phases increase the bordering lipid layer flexibility, promoting aggregate dissolution by a molecular- or surface-mobility-dependent mechanism. In this scenario, although aggregate particles that require appreciable energy to form are thermodynamically unstable, inter-particle repulsion may extend the life expectancy of such colloids to allow practical usage [26]. The nanosize and polydispersity affect NEs stability, rheology, appearance, color, texture, and shelf life, along with displaying improved pharmacological effects of the drugs [31]. Thus, NEs are promising drug delivery systems for pharmaceutical, cosmetic, and chemical industry applications.

#### 3.1.1. ALA-Nanoemulsion (BF-200 ALA)

ALA is a non-essential amino acid that occurs naturally in the human body, is involved in heme biosynthesis, and can be converted to protoporphyrin IX (PpIX) if delivered exogenously. PpIX induces phototoxicity and tissue destruction upon absorption of light of an appropriate wavelength and dose (40–200 J cm^−2^ at a wavelength of 610–635 nm) [32,33]. The properties of ALA-PpIX have been utilized in photodynamic diagnostics and PDT of cancer. This molecule is usually administered topically as a 20% cream on the skin and mucous membranes, but it has limited penetration due to its high hydrophilicity. Hence, ALA-induced PpIX formation is often restricted to superficial layers [34].

Maisch et al. evaluated a combination of nanoscale-lipid vesicle and gel formulations of the prodrug ALA (BF-200 ALA). The study utilized a porcine skin model to analyze the penetration of the BF-200 ALA (10% 5-ALA hydrochloride) versus a 16% MAL hydrochloride cream. At 8 and 12 h, the fluorescence signals of the metabolite PpIX were 4.8- and 5.0-fold higher, respectively, and almost two times greater depth compared to the MAL cream application [35]. The NE formulation stabilized the prodrug and enhanced its penetration through the *stratum corneum* (SC). However, MAL has shown to be less painful than BF-200-ALA, resulting in significantly lower treatment interruptions [36]. Besides, a systematic meta-analysis indicated that among BF-200 ALA, ALA-patch, methyl aminolevulinate (MAL), three modalities of imiquimod (3.75–5.0%), cryotherapy, 3% diclofenac in 2.5% hyaluronic acid, 0.5% 5-fluorouracil, and ingenol mebutate, the BF-200 ALA PDT was the most efficient treatment [37].

In 2010, Szeimies et al. treated 122 patients with actinic keratosis (AK) lesions with BF-200 ALA vs. placebo. After PDT with BF-200 ALA, the patient and lesion complete clearance rates reached 64% and 81%, respectively, achieving superior efficacy than placebo (11% and 22%, respectively) [30,38]. Subsequently, in 2012, Dirschka et al. evaluated 248 patients treated with BF-200 ALA PDT. The results revealed a higher patient complete clearance rate (78.2%) and an AK lesion complete clearance rate (90.4%) than placebo groups (17.1% and 37.1%). Additionally, BF-200 ALA displayed sustainable efficacy after a 1-year follow-up [39].

Another report by Dirschka et al. evaluated the efficacy of BF-200 ALA (7.8% ALA NE gel) vs. MAL (16% MAL cream) in the treatment of 52 patients in seven centers in Germany and Spain with mild-to-moderate AK utilizing daylight PDT (dPDT). The use of dPDT was intended to overcome a major PDT obstacle, especially the need for a specific illumination with a device that would assist in avoiding post-procedure pain. Impressively, 12 weeks after a single dPDT treatment, 79.8% of the AK lesions treated with BF-200 ALA gel and 76.5% of the lesions treated with MAL cream were completely cleared. Recurrence rates one year after treatment were 19.9% for lesions treated with BF-200 ALA and 31.6% with MAL. Moreover, AK’s dPDT with BF-200 ALA was well tolerated and non-inferior to MAL/dPDT regarding the efficacy and related pain, showing a trend towards higher efficacies after three months and significantly lower recurrence rates after 1-year follow-up. Therefore, dPDT reduced pain and simplified the procedure while maintaining high BF-200 ALA efficacy, showing to be a promising treatment for AK lesions [40].

Moreover, in a treatment study with basal cell carcinoma (BCC), 138 patients were treated with BF-200 ALA or MAL cream. The complete patient clearance rate for BF-200 ALA was 93.4%, compared to 91.8% for the MAL cream. For nodular BCC, 89.3% of the lesions were cleared with BF-200 ALA vs. 78.6% with MAL cream. Notably, lesion recurrence rates after 6 and 12 months were lower with BF-200 ALA (2.9% and 6.7%) in comparison with MAL cream (4.3% and 8.2%) [41]. Thus, NE ala formulations also demonstrated therapeutic superiority for BCC.

#### 3.1.2. Temoporfin (mTHPC)

Second-generation topical PSs used in PDT such as temoporfin (mTHPC) and zinc phthalocyanine (ZnPc) have also been incorporated into NE. Commercially known as Foscan^®^, mTHPC is widely used in systemic PDT for cancer therapy as a topical agent. In 2008, Primo et al. evaluated the photophysical and in vitro properties of biodegradable NE with mTHPC. The mTHPC diffusion flux was increased when this PS was incorporated into the NE [42]. In vitro assays showed an adequate profile for this system’s interaction in the different skin layers with an ideal time lag of 6 h. These parameters demonstrated that the NE could be potentially applied as a DDS for mTHPC in future PDT clinical applications involving topical skin cancer [30].

Furthermore, Primo et al. reported the synthesis and in vitro characterization of magnetic NE with mTHPC. Their results suggested that magnetic NE improves the penetration of mTHPC in skin layers leading to an adequate accumulation *in vivo*. The retention studies showed that the PS concentration in deep tissue layers was significantly higher in the presence of magnetic nanoparticles, making possible its topical application in skin cancer PDT protocols and hyperthermia activation in synergic procedures [43].

#### 3.1.3. Pc and Derivatives

The second-generation PS, ZnPc, was evaluated as a topical PDT agent to treat skin cancers. In 2008, Primo et al. developed a magnetic NE based on biodegradable surfactants. Preliminary in vitro assays indicated an excellent potential for synergic application in the topical release of ZnPc and excellent target tissue properties in PDT combined with hyperthermia activation [44]. ZnPc magnetic NE significantly increased the drug release in deeper porcine skin layers than the classical formulation in the absence of magnetic particles. This was suggested to be due to an increase in the DDS biocompatibility and affinity in the polar extracellular matrix of the skin. An increase in the drug partition inside the corneocytes could also be responsible for this action.

### 3.2. Lipid-Based Vesicular Systems

Phospholipid bilayers in a liquid crystalline state are desired for tDDS based on two broad liposomal formulations categories: conventional liposomes and novel liposomes such as the deformable/flexible/elastic lipid-based vesicles (Table 1). Conventional liposomes are artificial vesicles generally ranging in size from 20 to 1000 nm, composed mainly of an amphipathic phospholipid bilayer that may or may not contain cholesterol, surrounding an aqueous core [20]. They are considered biocompatible and safe for transdermal drug delivery of PSs (Figure 3A).

Hybrid lipoidal vesicular systems represent various liposomal systems that, apart from phospholipids, contain other additives yielding deformability or elasticity to the bilayers. Such deformable/flexible/elastic liposomal formulations have increased drug permeability through the skin [45]. Taken together, lipid-based vesicular systems have the potential to improve the topical delivery of PSs by simply incorporating new components into the liposomal formulation. 

Tertiary and quaternary mixtures of oil, water, and surfactants, the latter of which can be supplemented with a phospholipid, have a long history of applications on the skin. Such combinations form vesicular or sponge phases and often microemulsions. However, microemulsions are far less form-adaptable than hybrid lipoidal vesicles, as the latter not only has a flexible membrane but also can adjust their inner volume easily to vesicle shape changes [26].

#### 3.2.1. ALA and Derivatives

As previously indicated, ALA has problems of instability and low skin penetration. Different strategies to enhance ALA penetration for improved PDT results have been investigated. Liposomal delivery and synthesis of ALA esters are some of the most extensively studied systems [46]. Liposomes provide an enhanced ALA passage through the SC, resulting in more precise drug targeting into diseased cells. Liposome-encapsulated ALA transformation into PpIX is also higher in AK and BBCs than in normal adjacent skin [47].

Although the superiority of liposomal PSs was not initially demonstrated [48], in 2001, Pierre et al. proposed modified liposomes having lipid composition similar to the mammalian SC to optimize ALA transdermal delivery in skin cancers [49,50]. The liposomal vesicles containing 5.7% ALA showed higher skin retention (*p* < 0.05) on the epidermis with a decrease of skin permeation compared to an aqueous solution [50]. Likewise, ALA liposomal formulations resulted in prolonged ALA-induced PpIX accumulation, as well as better epidermal targeting [51]. Additionally, enhanced ALA penetration with hydrophobic derivatives of ALA (ALA-esters, ALA-amino acid derivatives, and ALA dendrimers) showed promising results in vitro [52]. However, although some ALA-esters yielded strong in vivo results, only the MAL formulation is currently used in cutaneous malignancies [34].

Approximately a decade later, Dyaderm^®^ (a non-invasive fluorescence imaging system) combined with ALA encapsulated liposomes were used to detect early NMSCs on the face, neck, chest, back, and hands of patients treated with UV light or heavy outdoor workers [53]. The diagnostic skin fluorescence system using liposomal encapsulated ALA offered the possibility for early detection of NMSC and was helpful at pre-clinical stages. After the commercial development of the liposomal ALA and its methyl ester (Metvix^®^) for PDT, it was found that liposome-entrapped precursors induced the expression of the proteolytic enzymes metalloproteinases (MMPs) in BCC [54]. The MMP-3 expression was blocked after using an MMP-3 inhibitor, suggesting a route to improve topical PDT effectiveness.

In parallel to the advance of conventional liposomal systems, research and development of deformable/flexible/elastic lipid vesicles as tDDS were considered for ALA-PDT towards skin cancer. Critical studies in applying ethosomes as ALA tDDS have also been developed [55,56]. As a lipid vehicle, ethosomes contain higher concentrations of ethanol and lipids. The presence of ethanol in the formulation allows drug solubilization (Table 1). It creates deformable lipid structures that can more easily promote a more profound skin permeation of both hydrophilic and lipophilic drugs (Figure 3C). Thus, they have superior drug skin retention and permeation enhancement as well as improved pharmaceutical properties, including stability at room temperature, high entrapment efficiency, and greater compatibility with the SC [57]. In vivo experiments by Fang et al. indicated that the penetration ability of ethosomes was superior to that of liposomes [58]. The enhancements of all the formulations ranged from 11- to 15-fold compared to that of the control (ALA in an aqueous solution) in PpIX intensity. Colorimetry detected no erythema in the irradiated skin, favoring ALA-ethosomes for clinical use as a tDDS in anticancer PDT [58]. Additionally, the ALA ethosomal formulation showed a 3.64-fold higher PpIX detection in mouse skin than the aqueous formulation [59].

More recently, cationic ALA-loaded ultra-deformable liposomes, also known as transfersomes (Figure 3B), showed higher stability and permeability than other tested liposomal formulations, enabling ALA delivery to deeper skin layers [49] (Table 1). Moreover, cationic transfersomes allowed higher retention of ALA in cells and improved the induction of PpIX, indicating better photosensitizing properties compared to other hybrid lipoidal vesicles.

In 2015, Bragagni et al. developed ALA-loaded niosomes, which are amphiphilic vesicles formed by synthetic non-ionic surfactants (Table 1). These lipid-based vesicles offer a series of potential advantages over liposomes, including more significant physical and chemical stability, longer shelf life, greater ease of production and storage, lower cost, and broader formulation versatility. Ex vivo permeation and penetration studies on excised human skin revealed that niosomal formulations were significantly more effective in improving ALA tDDS than aqueous drug solutions, leading to an 80% increase of drug permeation with 100% of the drug retained in the skin [60].

#### 3.2.2. Temoporfin (mTHPC)

Liposomal formulations have included a broad spectrum of second-generation PSs. Bendsoe et al. reported the human use of liposomal mTHPC in a topical gel formulation for PDT to treat NMSC [61]. Subsequently, Johansson et al. reported a novel, selective, and in-depth distribution of a liposomal mTHPC formulation applied for 4 and 6 h in a murine skin tumor model [62]. The results suggested that this PS formulation could be interesting for topical administration of mTHPC, decreasing the effects of extended skin photosensitivity associated with systemic mTHPC administration. Possible toxicities using a liposomal mTHPC were investigated, revealing no general adverse effects with cats after treatment [63]. However, a low degree of toxicity was demonstrated in 15% of the test subjects.

Although liposomal mTHPC has shown limited clinical value, this potent second-generation synthetic PS with hybrid lipoidal vesicles displayed outstanding performance as tDDS in anticancer PDT. To illustrate this, Dragicevic-Curic et al. repeated the topical application of mTHPC-loaded invasomes-vesicles, containing a mixture of terpenes or a single terpene with ethanol in addition to phospholipids (Table 1), onto the skin of mice bearing subcutaneously implanted human colorectal tumor HT29 followed by photoirradiation. The groups of mice treated with mTHPC-invasomes containing 1% of the terpene mixture before photoirradiation showed a significantly smaller (*p* < 0.05) tumor size increase compared to the control groups [64]. In vitro studies of mTHPC-invasomes and mTHPC-ethanolic solutions in two cancerous cell lines (HT29 and the epidermoid tumor cell line A431) were able to reduce the survival of these cell lines. Survival of only 16% of the A431 cells treated with mTHPC-invasomes revealed a promising tool for delivering mTHPC-PDT to cutaneous malignancies [65].

The same group showed that mTHPC-loaded ethosomes formulated with ethanol (3.3–20%, *w/v*) had higher in vitro percutaneous skin penetration than conventional liposomes using human abdominal skin mounted in Franz cells. The mTHPC-liposomes were of small particle size, small polydispersity index, negative surface charge, unilamellar or oligolamellar, and a spherical or oval shape. Remarkably, liposomes without ethanol delivered the lowest amount of mTHPC into the skin, while liposomes containing 20% ethanol showed the highest penetration. Thus, mTHPC-liposomes containing 20% ethanol could be a promising tool for delivering mTHPC to the skin, which could benefit the PDT of cutaneous malignant or non-malignant diseases [66].

Dragicevic-Curic et al. also evaluated the in vitro skin penetration of mTHPC-loaded neutral, anionic, and cationic transfersomes using human abdominal skin mounted in Franz diffusion cells (Figure 3B). Besides the effect of surface charge of transfersomes on skin penetration of mTHPC, its impact on physical properties (particle size, polydispersity index, lamellarity) and the physicochemical stability of vesicles were investigated (Table 1). From these mostly unilamellar and spherical vesicles, cationic transfersomes possessed the highest penetration enhancing ability (mTHPC -amount delivery to SC and deeper skin layers) than conventional liposomes, neutral, and anionic transfersomes. Regarding stability, contrasting to anionic transfersomes, neutral and cationic transfersomes were stable for 9 months at 4 °C. Thus, mTHPC-loaded cationic transfersomes showed the highest potential to be used in anticancer PDT [66].

#### 3.2.3. Pc and Derivatives

Pcs are among the more promising PS due to their intense absorbance in clinically effective red spectral regions (650–680 nm) and high singlet oxygen quantum yield. Aluminum (III) phthalocyanine tetrasulfonate (AlPcS4) has been heavily evaluated as a potential PS due to its ability to generate singlet oxygen. However, its hydrophilic and anionic nature hampers its transdermal delivery. Kassab et al. showed the efficacy of the hydrophilic tetra-anionic AlPcS4-loaded transfersomes (Trans-AlPcS4) as a novel technique for topical delivery in vitro (mammalian fibroblasts) and ex vivo (BALB/c mice dorsal skin). In vitro studies revealed a two-fold enhancement of the photocytotoxicity of Trans-AlPcS4 compared to free AlPcS4 dissolved in the culture medium. Ex vivo topical application on the dorsal skin of BALB/c mice revealed that both free AlPcS4 and Trans-AlPcS4 exhibited photosensitization towards mice skin [67].

As an essential second-generation PS, ZnPc has been evaluated in tDDSs to treat skin cancers with PDT. Bolfarini et al. prepared a magneto-liposome loaded with cucurbit [7] uril (CB [7]):ZnPc complex to improve the water solubility, dissolution, and bioavailability of this hydrophobic PS [68]. In vitro phototoxicity of both free and liposomal formulations was carried out on B16-F10 melanoma cells. The liposomal CB [7]:ZnPc showed excellent phototoxic effects for PDT applications. The cell survival ranged from 78.3% (±1.26) to 30.9% (±0.06) at the lowest and highest light dose, respectively [68].

Transfersomes have also been explored as topical DDS for ZnPc and the nitrosyl ruthenium complex [Ru (NH.NHq)(tpy)NO]^3+^ (RuNO) as a PS for co-generation of superoxide and nitric oxide as reactive species [69]. The transfersomes incorporating ZnPc and RuNO were phototoxic towards B16-F10 melanoma cells while having no dark toxicity. It was proposed that the novel topical transfersomes could be developed as a suitable tDDS for PDT [69].

Another important Pc, chloroaluminum phthalocyanine (ClAlPc), has been extensively explored in PDT against many cancer cell lines. However, ClAlPc is also known to be highly hydrophobic, requiring a tDDS association for clinical use [70]. Vilsinski et al. showed that ClAlPc diblock copolymer nanostructures presented highly efficient micellar morphology. In vitro tests using the Caco-2 human colon carcinoma cell line incubated with ClAlPc diblock nanomicelles at different Pc concentrations showed cellular damage after 30 min of LED radiation (663 nm, fluence of 1.62 μJ cm^-2^), but no cytotoxicity in the dark assays. Thus, the ClAlPc diblock copolymer produced promising nanomicelles suitable for incorporating the hydrophobic ClAlPc photosensitizer and subsequent use in the PDT [71]. Even more exciting, Almeida et al. developed a ClAlPc-loaded Nanostructured Lipid Carrier (NLC), a mixture of solid and liquid lipids dispersed in an aqueous surfactant solution, composed of 40% of oleic acid that resulted in approximately 99% reduction in BF16-F10 melanoma cell viability after light radiation. Usually, NLCs have great storage stability and high loading capacity [72]. Therefore, lipid-based nanocarriers can be valuable carrier systems for the Pc placement and the application of PDT in skin cancer treatment.

#### 3.2.4. Chlorophyll (CHL) and Derivatives

Some CHL derivatives and dyes belong to the second generation of PSs. They can absorb light at longer wavelengths and significantly reduce side effects of skin photosensitization due to rapid clearance compared to porphyrin-based PSs. The metallic chlorophyll derivatives have high absorption spectra at 400–430 nm and 650–670 nm regions and produce a high yield of superoxide [73]. The ROS production of these metal derivatives is ordered as such: Ferrous chlorophyllin (Fe-CHL) > Mg-CHL > Cu-CHL, resulting in Fe-CHL having the most robust photodynamic activity [74].

Noteworthy, although melanotic melanoma is a resistant tumor to various treatment strategies, including PDT due to melanin optical interference, depigmented melanomas were successfully treated with Fe-CHL liposomal formulations. Following depigmentation with phenylthiourea, Gomaa et al. observed a powerful combination of apoptosis and necrosis in melanoma cells treated with liposomal-Fe-CHL PDT [74] (Figure 4). Similarly, Rady et al. reported successful results of depigmented PDT-resistant melanoma treated with Fe-CHL trans-ethosomes, the combination of transfersome and ethosomes [70]. In vivo experiments achieved complete regression of small tumors after a single PDT session and regression of large tumors after two, with eight-month relapse free-survival [70].

To treat squamous cell carcinoma (SCC), Nasr et al. used Fe-CHL-loaded ethosomes and lipid-coated chitosan (PC/CHI) nanocarriers to enhance tDDS in PDT. Mouse skin ex vivo assays showed deeper penetration of ethosomes down to the dermis than PC/CHI nanocarriers, which were confined to the epidermis. However, they showed no significant difference in skin retention. Conversely, PC/CHI nanocarriers showed higher cytotoxicity in vitro against human SCC monolayers with no cytotoxic effects before laser exposure. Thus, both types of nanocarriers can be used for their potential treatment of SCC in PDT depending on the tumor depth and location in the skin [75].

## 4. Physical Approaches for Transdermal Delivery

### 4.1. Ablative Fractional Technology

The controlled disruption and ablation of the SC, the predominant barrier for topical drug delivery, can be achieved via microneedling, radiofrequency (RF), and lasers. Recently, the concept of using a laser to treat the skin has attracted increasing attention. Laser-assisted drug delivery (LADD) involves controlled, selective destruction of the epidermis and dermis to allow for penetration and absorption of topical medications and large drug molecules. Challenges in predicting LADD’s efficacy and safety include: the unpredictability of drug dosing and possible systemic toxicity, variability in absorption, induction of localized and systemic hypersensitivity, and inconsistencies in treatment protocol [76] (Figure 5).

Lasers of different wavelengths and types have been employed to increase drug permeation. These include the ruby, erbium:yttrium-aluminum-garnet (Er:YAG), neodymium-doped yttrium-aluminum-garnet, and CO_2_ lasers [77]. Fractional modality is a novel concept for promoting topical/transdermal drug delivery [78]. The laser helps enhance the permeation because of its capacity to produce microscopic ablated vertical channels. However, LADD parameters need to be adjusted to the patient, skin condition, location, and drug employed. LADD has been used with various topical products, including PS ALA for AKs and NMSCs. Generally, LADD is a promising technique that enhances topical molecules’ absorption while adding to the laser’s synergic effect [77].

#### ALA and Its Derivatives

Shen et al. were among the first groups to study the in vivo kinetics of PpIX generation after topical ALA application enhanced by an Er:YAG laser. The enhancement ratios of PpIX with laser-treated murine skin ranged from 1.7 to 4.9-times compared to the control group. The PpIX was more concentrated in superficial epidermal layers with the control group than that of the laser-treated group. Furthermore, the barrier properties of the laser-treated skin rapidly recovered within three days. Thus, pretreatment of the skin using an Er:YAG laser showed to be useful in increasing the amount of PpIX within skin tumor cells [79].

Additionally, Haedersdal et al. evaluated PS drug delivery of ALA and MAL by a CO_2_ ablative fractional laser (AFXL). They established that AFXL increased topical uptake of these PSs using stacked single 91.6 mJ pulses of 3 milliseconds, followed by topical MAL application for 3 h (Metvix^®^). This condition generated approximately 3 mm apart laser microchannels in the skin and consequently a more homogeneous distribution of the photobleached porphyrins’ fraction throughout the skin [80,81]. In a comparative study of the kinetics and biodistribution of ALA- and MAL-induced porphyrins on intact versus AFXL-exposed swine skin, the latter showed a considerably enhanced signal of the porphyrin fluorescence of both PSs (*p* < 0.05). On AFXL-treated skin, MAL briefly generated a higher fraction of photobleached than ALA. However, ALA induced a higher photobleached fraction than MAL over time. Additionally, a higher fraction of photobleached porphyrins were observed in hair follicle epithelium for ALA compared to MAL, implying that AFXL-ALA favors targeting deeper structures [82].

More recently, Paasch et al. evaluated CO_2_ AFXL-LADD combined with indoor daylight (IDL) ALA-PDT for effectiveness and safety to treat skin field cancerization associated with AK. Impressively, all 46 patients showed remission (complete: 71.7%, partial: 28.3%), suggesting that AFXL-LADD combined with IDL-PDT is an exceptionally effective treatment. Nevertheless, the high pain scores associated with this combined approach may prove to be a limiting factor [83].

Other modalities of AFXL to improve tDDS, such as fractional RF and thermomechanical fractional injury (TMFI), have been evaluated. Park et al. demonstrated that fractional RF with sonophoresis effectively enhanced ALA penetration in swine skin. Pre-fractional RF followed by post-treatment with sonophoresis was considered a promising therapeutic combination for ALA-PDT to enhance ALA uptake [84]. Additionally, Shavit et al. evaluated the efficacy of pretreatment by TMFI (Tixel^®^, Novoxel^®^, Israel) at low-energy settings to increase the permeability of the skin to four topical PS preparations, especially: 20% ALA gel prepared in a good manufacturing practice-certified pharmacy (Super-Pharm Professional, Israel), 10% ALA microemulsion gel (Ameluz^®^, Biofrontera Bioscience GmbH, Leverkusen, Germany), 16.8% MAL cream (Metvix^®^, Galderma, Lausanne, Switzerland), and 20% ALA hydroalcoholic solution (Levulan Kerastick^®^, DUSA Pharmaceuticals, Inc., Wilmington, MA, USA). Pretreatment with low-energy TMFI at a pulse duration of 6 milliseconds increased the percutaneous permeation of ALA when the 20% gel was used. Incredibly, after 2 and 3 h, the TMFI-treated sites exhibited an increased hourly rate of PpIX fluorescence intensity, which was 156–176% higher than the control (*p* ≤ 0.004). Thus, TMFI seems to be a powerful method to enhance the transdermal drug delivery of ALA and its derivatives. Additionally, the formulation’s characteristics significantly influence TMFI pretreatment adjutancy [85].

However, although the positive results of LADD-ALA-PDT were observed, a comparative study by Chen et al. to evaluate ALA-PDT revealed that plum-blossom needling (a method of shallow insertion of multiple needles into the skin) had more broad diffusion of ALA than the CO_2_ AFXL while having a similar clinical effect at a much lower cost. A clinical trial also revealed that the surface fluorescence intensity was stronger in needle-pretreated-lesion than in laser-pretreated-lesion. It appears that plum-blossom needling treatment may be clinically superior for enhancing ALA-tDDS and other topical skin medications [86].

### 4.2. Microneedling

The parenteral administration of near-IR preformed PSs suffers from low selectivity and may result in prolonged skin photosensitivity. MNs can provide localized drug delivery to skin lesions and overcome the limitations of delivering into the dermal layer. In superficial cancer treatments, topical drug administration faces severely low transfer efficiency (Figure 6). MN-based systems have achieved excellent administration capabilities and have been tested for pre-clinical PDT [87].

Donnelly et al. recently reported on the uses of MNs with ALA-tDDS [88,89,90,91,92,93,94]. They employed silicon MNs to overcome the limitation of the low tissue penetration of ALA. Both in vitro and in vivo results revealed that the MNs increased the skin penetration of the PS and also enhanced the PpIX production [88]. Additionally, Zhao et al. used sodium hyaluronate to create fast-dissolving MNs patches [95]. Even though the injection dose was relatively low, the transdermal pathway achieved a much better tumor inhibition rate (66–97%) than direct injection. Conversely, Jain et al. coated ALA on solid MNs and evaluated them in a porcine skin model. In comparison with conventional cream formulation, the delivery efficacy of these MNs was 3.2-fold higher, with PpIX being generated at least three times greater amounts (~480 μm), and with better anti-tumor effects [96].

Donnelly et al. also compared the delivery performance of dissolving- versus hydrogel-forming MNs loaded with ALA and meso-tetra (N-methyl-4-pyridyl) porphine tetra tosylate (TMP). Microneedling significantly enhanced transdermal delivery of both ALA and TMP in vitro. The MN hydrogel-forming system was comparable with the MN dissolving system for ALA delivery (~3000 nmol/cm^2^ over 6h), however superior for delivery of the much larger TMP molecule (~14 nmol/cm^2^ over 24h, compared to 0.15 nmol/cm^2^) [91]. Thus, these results have opened the potential for investigating microneedling with many other PSs.

#### 4.2.1. Bacteriochlorin

Intradermal delivery of the preformed near-IR PS: 5,10,15,20-tetrakis (2,6-difluoro-3-N-methyl-sulfamoylphenyl bacteriochlorin (Redaporfin^TM^) using dissolving MNs was successful in vitro and in vivo to treat nodular BCC. MNs demonstrated complete dissolution 30 min after topical application and revealed sufficient mechanical strength to penetrate 450 μm depth of the skin. In vitro studies illustrated that the drug delivery and detection was 5 mm in-depth of the skin. In vivo biodistribution studies in athymic nude mice showed both fast initial release with localized drug delivery. The MN-treated mice showed a progressive decrease in the PS at the application site over 7 days. However, most of the skin surface showed fluorescence levels comparable to those seen in the negative control group. These results suggested beneficial effects for polymeric MN arrays in minimally invasive intradermal delivery to enhance PDT in deep skin lesions [97].

#### 4.2.2. Pc and Derivatives

MN applications with Pcs have also shown encouraging results. Tham et al. developed a mesoporous nanovehicle with dual loading of PSs and clinically relevant drugs for combination therapy while utilizing MN technology to facilitate their penetration into the deep skin tissue [98]. Impressively, the organo-silica matrix dramatically increased the quantum yield and photostability of Pcs. In ex vivo studies, porcine skin fluorescence imaging demonstrated that MNs could facilitate the penetration of the nano-vehicles across the epidermis layer of skin to reach deep-seated melanoma sites. After one hour of the topical delivery of Pc-nanoparticles or free Pc, the signal on MN-treated samples increased significantly. The amount of Pc that permeated the skin was 27.2% and 63.1% without and with the MN, respectively. Remarkably, there was minimal skin penetration of free Pc, regardless of the MN treatment. Besides, the nanoparticles’ mesopores were further loaded with small-molecule inhibitors, such as dabrafenib and trametinib, which target the hyperactive mitogen-activated protein kinase (MAPK) pathway for melanoma treatment. NIR-irradiated drug-loaded nanovehicle revealed a synergistic killing effect on melanoma cells through ROS and caspase-activated apoptosis. Tumor regression studies on a xenografted melanoma mouse model highlighted superior therapeutic efficacy of the nanovehicle through combinational PDT and targeted therapy as a promising therapeutic option for malignant melanoma [99].

More recently, Shi et al. developed an innovative MN-dissolving platform for combining PDT and immunotherapy via controlled co-delivery of a ZnPc and checkpoint inhibitor anti-CTLA4 antibodies. This combination generated synergistic reinforcement outcomes while reducing side effects. This co-loading carrier is effectively aggregated around superficial tumors by MN-tDDS. In vivo studies using a mouse 4T1 breast cancer model with PDT initially exerted tumor cytotoxicity and triggered the immune responses, synergistically aiding immunotherapy. The possible mechanism and systemic effects of the combined therapy demonstrated that this co-administration platform could be a promising tool for focal cancer treatment [100].

## 5. Conclusions

PDT for cutaneous malignancies has seen a multitude of technological advancements, which allows for an increase in efficacy and expansion in applications. Impressive clinical outcomes of this treatment method on more aggressive skin cancers, such as malignant melanoma, have recently been reported. Beyond ALA and its derivatives, the use of second-generation PSs, such as mTHPC and Pcs, as well as third-generation have seen an explosion in applications. Novel transdermal delivery systems have been shown to mitigate local pain and inflammation as well as lead to increased practicality through the use of daylight irradiation, increasing treatment adherence.

The use of physical enhancers has shown promise to incorporate heightened formulations such as liposomes, hybrid lipoidal vesicles, and NEs. Additionally, combinations with other advanced therapeutic agents such as chemotherapeutics, immunomodulators, and targeted therapy have demonstrated success. These advances allow future perspectives in specific molecular targeting and immunomodulation of the tumor microenvironment, which opens the possibility for the development of a transdermal anticancer vaccine with a dual role of (1) destroying cancer cells upon photoactivation exposing tumoral antigens, and consequently (2) promoting a durable anticancer immune response, not only locally, but also in distant metastasis.

## Figures and Tables

**Figure 1 pharmaceuticals-14-00772-f001:**
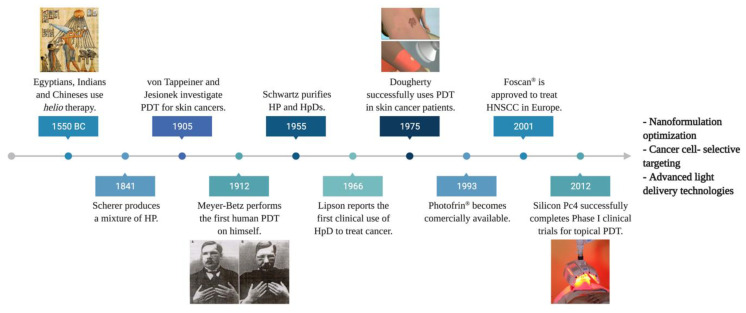
The developmental timeline of photodynamic therapy through the centuries. Abbr.: hematoporphyrin (HP), photodynamic therapy (PDT), hematoporphyrin derivative (HpD), head-and-neck squamous carcinoma (HNSC), phthalocyanine (Pc).

**Figure 2 pharmaceuticals-14-00772-f002:**
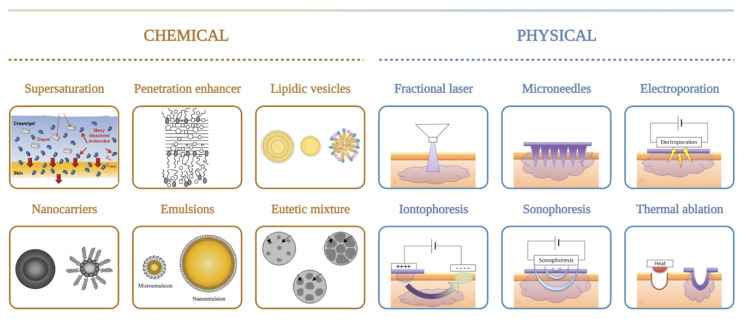
Current chemical and physical strategies to enhance transdermal drug delivery of photodynamic therapy.

**Figure 3 pharmaceuticals-14-00772-f003:**
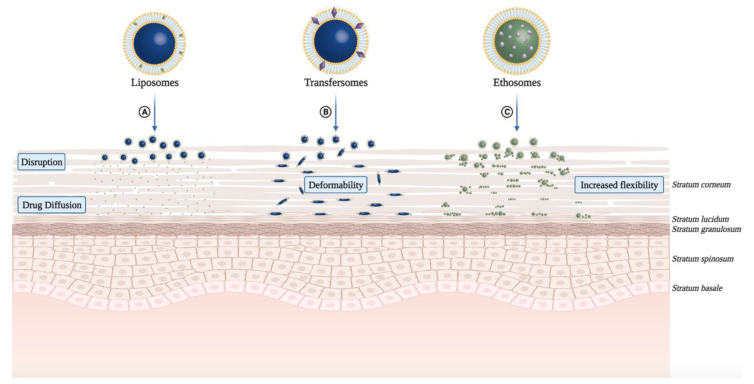
Mechanisms of skin permeation of lipid-based nano-vesicular systems for transdermal drug delivery in photodynamic therapy. (**A**) Conventional liposomes are superficially retained in the *stratum corneum*, disrupting and releasing the drug, which continues the epidermal penetration through diffusion. (**B**) Transfersomes have edge activators (surfactants) in their composition, which confer deformability, allowing for deeper drug release in the skin. (**C**) Ethosomes have a higher concentration of ethanol in their composition, increasing their flexibility and allowing for deeper drug release in the skin.

**Figure 4 pharmaceuticals-14-00772-f004:**
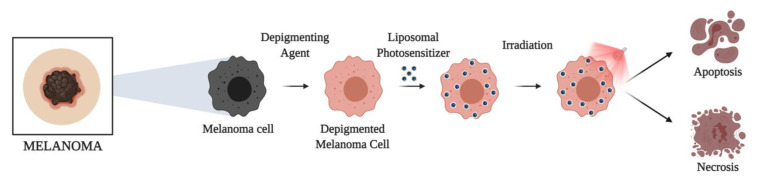
Enhanced photodynamic therapy combining neoadjuvant topical depigmentation and liposomal photosensitizing drug delivery to treat resistant melanotic melanoma.

**Figure 5 pharmaceuticals-14-00772-f005:**
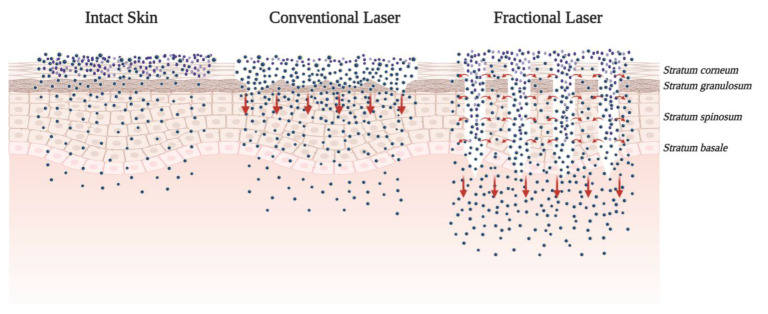
Mechanism of action of fractional laser-assisted photosensitizing drug delivery systems for PDT enhancement.

**Figure 6 pharmaceuticals-14-00772-f006:**
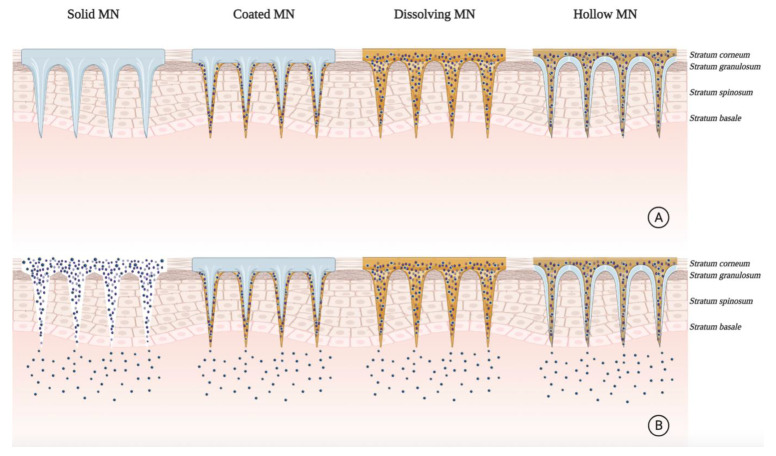
Mechanism of action of current microneedle (MN)-based systems for photosensitizing drug delivery enhancement in photodynamic therapy. (**A**) Graphical representation of current microneedling systems applied to the skin. (**B**) Graphical representation of skin permeation of photosensitizer microneedle-based platforms.

**Table 1 pharmaceuticals-14-00772-t001:** Current lipid-based nano- and micro-vesicular systems in use for topical and transdermal drug delivery in photodynamic therapy. Abbr.: 5-aminolevulinic acid (ALA), meso-tetra (hydroxyphenyl)-chlorin (mTHPC), benzoporphyrin derivative monoacid ring A (BPD-MA), zinc phthalocyanine (ZnPc).

Types	Nano-/Micro-Emulsions	Conventional Liposomes	Niosomes	Transfersomes	Ethosomes	Invasomes
Structure	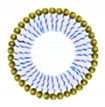	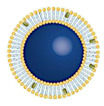	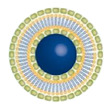	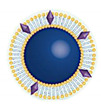	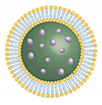	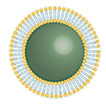
Timeline	1961	1965	1970	1991	1997	2004
Composition	Oil or main oleaginous ingredient Surfactant/emulsifier Water	Phospholipids and cholesterol	Non-ionic surfactants 5–10% Other additives Buffer QS	Lipid 5% *w/v* Edge activator 1% *w/v* Buffer QS	Phospholipid 5–10% *w/w* Ethanol 20-50% *w/w* Buffer QS	Soy-phosphatidylcholine 10% *w/w* and Lyso soy-phosphatidylcholine 0.7% Ethanol 10% *w/w* Terpenes 1% *w/w* Buffer QS
Characteristics	A liquid colloidal dispersion system which is kinetically stable, with a droplet size <100 nm	Microscopic spheres	Microscopic spheres	Ultra-deformable liposome	Elastic liposome	Ultraflexible and Elastic liposome
Flexibility	-	Rigid	Rigid	High deformability due to edge activator (surfactant)	High deformability and elasticity due to ethanol	High deformability and elasticity
Permeation Mechanism	Diffusion/Fusion/Lipolysis	Diffusion/Fusion/Lipolysis	Diffusion/Fusion/Lipolysis	Deformation	Lipid perturbation	Deformation and Lipid perturbation
Trademarks of topical photosensitizers	5-ALA Ameluz^®^	Lipoxala^®^ spray, mTHPC Foslip^®^, BPD-MA Visudyne^®^, ZnPc CGP55847^®^ BPD-MA	-	mTHPC Fospeg^®^	-	-

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
