# Peer review of "Micro- and Nano-Based Transdermal Delivery Systems of Photosensitizing Drugs for the Treatment of Cutaneous Malignancies"

_pharmaceuticals, 2021, doi:10.3390/ph14080772_

Round 1

Reviewer 1 Report

The manuscript pharmaceuticals-1323078 is a review article focused on the use of photodynamic therapy (PDT) for cutaneous pre-cancerous lesions and skin cancers. This review is well organized, easy to read, and includes precise updates on the latest progress made in transdermal drug delivery of PDT agents, exploring diverse chemical and physical approaches for transdermal delivery. The references are appropriate and updated; Figures and Table are clear and professional, and the topic fits within the scope of Pharmaceuticals. Therefore, I would recommend the publication of this work after addressing the following minor comments:  

  1. In the Introduction: page 2, line 64. Please add references.
  2. In the Introduction: page 3, lines 67-81. A very interesting historically overview of the use of phototherapy is given. Besides being described in the following sections, the commercially available Photofrin® and Foscan®, and also the most recent Phase I clinical trial for topical PDT are well shown in Figure 1 but are not referred in the text. Thus, I would recommend adding this information, at the end of page 4, line 83.
  3. In the Introduction: page 2, lines 77-80. Reference [6] should be reference [7] and reference [7] should be reference [6]. Please change in the references list.  
  4. In page 5, line 119; page 10, line 244; page 24, line 536; and in page 25, line 567. Please add supporting references.
  5. In Figure 3, there is a missing “R” in the word “Transfersomes”. Please correct it.
  6. In Figure 3 and in Figure 6, please delete the “A.” after PDT.
  7. I would suggest adding “(Figure 3B)” in page 13, line 314.
  8. In page 17, lines 400 and 404, there is a missing space before the corresponding reference.
  9. In page 19, line 435, please confirm if Table 1 is complete. At least in the pdf file, the last column is not readable.
  10. Finally, I would like to say that I found very interesting your future perspective on the development of a transdermal anticancer vaccine.

Author Response

The authors wish to express their appreciation for all the valuable suggestions. Below is a list of how we have addressed them all.

1. In the Introduction: page 2, line 64. Please add references.

The following references have been added:

[3] M.C.S. Vallejo, N.M.M. Moura, M.A. Ferreira Faustino, A. Almeida, I. Gonçalves, V.V. Serra, M.G.P.M.S. Neves, An Insight into the Role of Non-Porphyrinoid Photosensitizers for Skin Wound Healing, International Journal of Molecular Sciences 22(1) (2020) 234.

[4] M.C.S. Vallejo, N.M.M. Moura, A.T.P.C. Gomes, A.S.M. Joaquinito, M.A.F. Faustino, A. Almeida, I. Gonçalves, V.V. Serra, M.G.P.M.S. Neves, The Role of Porphyrinoid Photosensitizers for Skin Wound Healing, International Journal of Molecular Sciences 22(8) (2021) 4121.

[5] D. Yang, S. Lei, K. Pan, T. Chen, J. Lin, G. Ni, J. Liu, X. Zeng, Q. Chen, H. Dan, Application of photodynamic therapy in immune-related diseases, Photodiagnosis and Photodynamic Therapy 34 (2021) 102318.

[6] E.G. Mik, Measuring Mitochondrial Oxygen Tension: From Basic Principles to Application in Humans, Anesthesia & Analgesia 117(4) (2013) 834-846.

2. In the Introduction: page 3, lines 67-81. A very interesting historically overview of the use of phototherapy is given. Besides being described in the following sections, the commercially available Photofrin® and Foscan®, and also the most recent Phase I clinical trial for topical PDT are well shown in Figure 1 but are not referred in the text. Thus, I would recommend adding this information, at the end of page 4, line 83.

The following text has been added within the manuscript:

“Further, Dougherty et al. introduced PDT in the 1970s by observing complete mammary tumor remission in vivo using HpD combined with the red light These findings led to the first approval of PDT drug, Photofrin®, to treat bladder cancer in Canada in 1993 ”

3. In the Introduction: page 2, lines 77-80. Reference [6] should be reference [7] and reference [7] should be reference [6]. Please change in the references list.

These references were changed accordingly. 

4. In page 5, line 119; page 10, line 244; page 24, line 536; and in page 25, line 567. Please add supporting references.

The following references have been added:

[23] S. Supe, P. Takudage, Methods for evaluating penetration of drug into the skin: A review, Skin Research and Technology 27(3) (2021) 299-308.

[24] R.K. Keservani, S. Bandopadhyay, N. Bandyopadhyay, A.K. Sharma, Design and fabrication of transdermal/skin drug-delivery system, Drug Delivery Systems, Elsevier 2020, pp. 131-178.

[42] F.L. Primo, M.V.L.B. Bentley, A.C. Tedesco, Photophysical studies and in vitro skin permeation/retention of Foscan/nanoemulsion (NE) applicable to photodynamic therapy skin cancer treatment, Journal of nanoscience and nanotechnology 8(1) (2008) 340-347.

[43] F.L. Primo, L. Michieleto, M.A.M. Rodrigues, P.P. Macaroff, P.C. Morais, Z.G.M. Lacava, M.V.L.B. Bentley, A.C. Tedesco, Magnetic nanoemulsions as drug delivery system for Foscan®: Skin permeation and retention in vitro assays for topical application in photodynamic therapy (PDT) of skin cancer, Journal of Magnetism and Magnetic Materials 311(1) (2007) 354-357.

[91] R.F. Donnelly, D.I.J. Morrow, M.T.C. Mccrudden, A.Z. Alkilani, E.M. Vicente-Pérez, C. O'Mahony, P. González-Vázquez, P.A. Mccarron, A.D. Woolfson, Hydrogel-Forming and Dissolving Microneedles for Enhanced Delivery of Photosensitizers and Precursors, Photochemistry and Photobiology 90(3) (2014) 641-647.

[99] V. Alimardani, S.S. Abolmaali, G. Yousefi, Z. Rahiminezhad, M. Abedi, A. Tamaddon, S. Ahadian, Microneedle Arrays Combined with Nanomedicine Approaches for Transdermal Delivery of Therapeutics, Journal of Clinical Medicine 10(2) (2021) 181.

5. In Figure 3, there is a missing “R” in the word “Transfersomes”. Please correct it.

We have corrected and added an updated image.

6. In Figure 3 and in Figure 6, please delete the “A.” after PDT.

A.” is part of the figure caption, but for clarity we have replaced with A)

7. I would suggest adding “(Figure 3B)” in page 13, line 314.

We have included this within the text as suggested.

8. In page 17, lines 400 and 404, there is a missing space before the corresponding reference.

This has been corrected and we have also corrected others which we found on our re-read.

9. In page 19, line 435, please confirm if Table 1 is complete. At least in the pdf file, the last column is not readable.

Table 1 is complete in our document (paragraph space removed).

10. Finally, I would like to say that I found very interesting your future perspective on the development of a transdermal anticancer vaccine.

We were please to read this.

Reviewer 2 Report

The manuscript reviews the progresses on transdermal delivery of photosensitizers for photodynamic therapy. The authors summarizes the chemical and physical approaches of micro- & nano systems for the delivery of therapeutic agents across the skin barriers. The topic is interesting and well-suited for the general readership of the journal. The manuscript is recommended for publication after minor revision. 1. PDT is mainly for cancer treatment. And the content of the text is also mainly related to cancer therapy. Therefore the title of manuscript is overpraised and should be modified by the authors. 2. It is wondering whether the figures are produced by the authors or come from literatures. If it is in the latter case, citations are necessary in the captions. 3. As a review paper, more citations from the recent years are definitely necessary, especially from year of 2020 and 2021. 4. Typos should be avoid in the text. Just as an example, page number or article number is missing in reference 11, 15, 40 and 44.

Author Response

The authors wish to express their appreciation for all the valuable suggestions. Below is a list of how we have addressed them all.

1. PDT is mainly for cancer treatment. And the content of the text is also mainly related to cancer therapy. Therefore, the title of manuscript is overpraised and should be modified by the authors.

We have modified the title to now read as: “Micro- & nano-based transdermal delivery systems of photosensitizing drugs for the treatment of cutaneous malignancies

2. It is wondering whether the figures are produced by the authors or come from literatures. If it is in the latter case, citations are necessary in the captions.

All figures are produced by the authors.

3. As a review paper, more citations from the recent years are necessary, especially from year of 2020 and 2021.

We have included the below additional citations from 2020 and 2021:

[3] M.C.S. Vallejo, N.M.M. Moura, M.A. Ferreira Faustino, A. Almeida, I. Gonçalves, V.V. Serra, M.G.P.M.S. Neves, An Insight into the Role of Non-Porphyrinoid Photosensitizers for Skin Wound Healing, International Journal of Molecular Sciences 22(1) (2020) 234.

[4] M.C.S. Vallejo, N.M.M. Moura, A.T.P.C. Gomes, A.S.M. Joaquinito, M.A.F. Faustino, A. Almeida, I. Gonçalves, V.V. Serra, M.G.P.M.S. Neves, The Role of Porphyrinoid Photosensitizers for Skin Wound Healing, International Journal of Molecular Sciences 22(8) (2021) 4121.

[5] D. Yang, S. Lei, K. Pan, T. Chen, J. Lin, G. Ni, J. Liu, X. Zeng, Q. Chen, H. Dan, Application of photodynamic therapy in immune-related diseases, Photodiagnosis and Photodynamic Therapy 34 (2021) 102318.

[7] G. Gunaydin, M.E. Gedik, S. Ayan, Photodynamic Therapy—Current Limitations and Novel Approaches, Frontiers in Chemistry 9 (2021) 691697.

[23] S. Supe, P. Takudage, Methods for evaluating penetration of drug into the skin: A review, Skin Research and Technology 27(3) (2021) 299-308.

[24] R.K. Keservani, S. Bandopadhyay, N. Bandyopadhyay, A.K. Sharma, Design and fabrication of transdermal/skin drug-delivery system, Drug Delivery Systems, Elsevier 2020, pp. 131-178.

[99] V. Alimardani, S.S. Abolmaali, G. Yousefi, Z. Rahiminezhad, M. Abedi, A. Tamaddon, S. Ahadian, Microneedle Arrays Combined with Nanomedicine Approaches for Transdermal Delivery of Therapeutics, Journal of Clinical Medicine 10(2) (2021) 181.

4. Typos should be avoided in the text. Just as an example, page number or article number is missing in reference 11, 15, 40 and 44.

Thank you. We have carefully edited these and made sure any others were also formatted correctly.